



# Technical note: Gas Chromatography vs. Mid-Infrared Laser Absorption Spectroscopy: A comparison of methods for measuring greenhouse gas fluxes from arable soils

Wolfgang Aumer*[1,2], Morten Möller*[3], Carolyn-Monika Görres[1], Christian Eckhardt[4], Tobias Karl David Weber[5], Carolina Bilibio[5], Christian Bruns[3], Andreas Gattinger[6], Maria Renate Finckh[2], Claudia Kammann[1]

[1] Department of Applied Ecology, Hochschule Geisenheim University, Von-Lade-Str.1, D-65366 Geisenheim, Germany
[2] Section of Ecological Plant Protection, University of Kassel, Nordbahnhofstr. 1a, D-37213 Witzenhausen, Germany
[3] Section of Organic Farming and Cropping Systems, University of Kassel, Nordbahnhofstr. 1a, D-37213 Witzenhausen, Germany
[4] Institute for Plant Ecology, Giessen University, Heinrich-Buff-Ring 26, D-35392 Gießen, Germany
[5] Section of Soil Science, University of Kassel, Nordbahnhofstr. 1a, D-37213 Witzenhausen, Germany
[6] Chair of Organic Farming, Institute of Agronomy and Plant Breeding II, Justus-Liebig University Giessen (JLU), Gießen, Germany

*Wolfgang Aumer and Morten Möller contributed equally and are considered co-first authors.
*Correspondence to*: Wolfgang Aumer (wolfgang.aumer@hs-gm.de) and Morten Möller (morten.moeller@uni-kassel.de)

## Abstract

For the study of soil-atmosphere exchange of green-house gases, a commonly adopted method is to monitor the change of gas concentrations in closed chambers. Accurate determination of $CO_2$, $CH_4$, and $N_2O$ concentrations is therefore essential for reliable flux estimations. This study compares two techniques to determine these gas concentrations: Gas Chromatography (GC) and mid-infrared laser absorption spectroscopy (LAS). We compared both techniques by carrying out simultaneous chamber measurements under field conditions on two separate days covering a range of fluxes. The GC method involved syringe sampling into gas-tight vials and subsequent laboratory analysis. In contrast to that, a LAS analyzer was directly connected to the chambers (tubing system) and thus enabled real-time, high-temporal resolution data. We calculated gas fluxes based on GC- and LAS-derived concentration measurements, using seven distinct flux calculation setups, including systematic variations in chamber enclosure times (30, 20 and 10 min) for LAS data. Across both measurement days, the comparison resulted in a high level of agreement for determined $CO_2$ fluxes with a normalized Root Mean Square Error (nRMSE): 5.79 – 16.70 %. A high level of agreement between the methods was also observed for $N_2O$ fluxes (nRMSE: 14.63 – 24.64 %). In contrast, there was a comparatively low agreement between methods for $CH_4$ fluxes (nRMSE: 88.42 – 94.54 %). $N_2O$ and $CH_4$ fluxes highlighted the superior precision of LAS, as it detected significant fluxes (> minimum detectable flux) that were not significant with GC. For $CH_4$ this explains the low agreement between methods regarding arable soils that are dominated by (low) $CH_4$-consumption fluxes.



## 1. Introduction

Various methods are available to determine greenhouse gas (GHG) fluxes at the atmosphere-soil interface. A common and versatile approach to determine GHG fluxes like carbon dioxide ($CO_2$), nitrous oxide ($N_2O$), and methane ($CH_4$) in field experiments is the closed chamber method, where the soil surface is temporarily covered by a chamber attached to a collar that is permanently anchored in the soil to ensure an air-tight seal when the chamber is placed on top of it (Livingston and Hutchinson, 1995). The gas flux rate is then calculated based on the measured change in gas concentration within the chamber

headspace over time, using either a linear or non-linear model (Lundegardh, 1927; Hutchinson and Mosier, 1981; Yu and Yao, 2017; Maier et al., 2022). Ultimately, numerous analytical techniques are available for measuring these GHG concentrations, with decisions influenced by specific research questions and logistical feasibility. Gas concentrations required for flux calculations can be determined through real-time monitoring using online analyzers, or by collecting gas samples with syringes for subsequent analysis via gas chromatography (GC), followed by peak integration and calculation of gas concentrations to

estimate fluxes. In both cases, flux calculation procedures demand variables according to the ideal gas law such as chamber volume, temperature and local atmospheric air pressure and the resulting flux rates are subsequently referenced to the covered soil surface (Livingston and Hutchinson, 1995).

Besides the chamber design, the choice of the gas analytical technique can be an influential factor estimating GHG fluxes in the ecosystems under investigation. For example, a comparison between cavity ring-down spectroscopy (CRDS) and GC

(Christiansen et al., 2015a) showed that CRDS resulted in higher calculated $CO_2$ fluxes, provided comparable results for $N_2O$, and was significantly more sensitive for $CH_4$ fluxes compared to GC. The authors concluded, that both CRDS and GC were equally effective in capturing treatment effects for $CO_2$ and $N_2O$ in laboratory and field settings, whereas this was not the case for $CH_4$. Zheng et al. (2008) pointed out, that measuring $N_2O$ concentrations using GC with an Electron Capture Detector (ECD), the common practice using nitrogen ($N_2$) as a carrier gas can lead to overestimated $N_2O$ emissions, especially in the

presence of $CO_2$ or from weak sources ($< 200$ µg N m$^{-2}$ h$^{-1}$). To address this issue, they suggest using alternative methods, for instance chemical removal of $CO_2$. Although the GC has commonly been used in the past, new modern online analyzers based e.g. on mid-infrared laser absorption spectroscopy (LAS) allow real-time concentration measurements within the chamber where analytical values are immediately displayed, as opposed to the GC analysis which does not allow this in most cases due to the time gap between taking samples and subsequently analyzing them; exceptions being a GC mounted in a van or container

in the field to a multiplexer plus automated chamber system (e.g. Flessa et al., 2002; Yao et al., 2009). An immediate display of the measured concentration, e.g. with an LAS analyzer, provides the advantage of immediate detection of methodical set-up issues, such as leaks in chambers or tubing, enabling corrections or repetitions as necessary. Furthermore, LAS often offer higher analytical precision, thus enabling the detection of smaller flux rates and more precise measurements within the low concentration range, which means that the minimum detectable flux (MDF) is reduced notably (Christiansen et al., 2015a;

Nickerson, 2016).



Several components of the closed chamber method significantly affect MDF, including chamber size, closure duration, sampling frequency during closure (i.e., periodicity), and the analytical precision of gas analysis technique. Substantial advancements in analytical precision and temporal resolution have led to markedly improved accuracy of GHG flux measurements, enabling shorter chamber closure times (Brümmer et al., 2017; Johannesson et al., 2024) and thereby minimizing disturbance and flux gradient alteration (Livingston and Hutchinson, 1995; Maier et al., 2022). This study aims to serve as a pilot study on the comparability of the classic GC method and fast developing LAS analyzers. A challenge of adoption of new technology is also that it changes the inherent measurement uncertainties, thus, knowledge about the impact of the gas analytical method on the gas fluxes are required (Cowan et al., 2025; Kong et al., 2025). Thus, the objective is to assess how closely the GHG flux values from both methods align and to evaluate the limitations of each method in terms of measurement accuracy. For this purpose, we conducted closed chamber measurements using simultaneously static (GC) and dynamic (LAS) approaches under field conditions and calculated the corresponding MDF.

## 2. Material and methods

### 2.1. Site description

The measurements were carried out in a long-term field experiment (LTE), initiated in 2010, which is well described in Bilibio et al. (2025)Klicken oder tippen Sie hier, um Text einzugeben.. The LTE is located near Neu-Eichenberg, Hesse, Germany (223 m asl., 51°22'N 9°54'E), within an organic research station operated by the University of Kassel which is further described in Leisch et al. (2025). The geological formation (Keuper) is covered by a loess layer up to a thickness of 1.8 m. The soil, characterized as a silty loam, is classified as a Luvisol (Obalum et al., 2019). It consists of 13 % clay, 84 % silt, and 3 % sand, with an organic matter content of 2 % (Schmidt et al., 2017). Over a 30-year period (1991 to 2020), the average annual temperature recorded was 9.3 °C, accompanied by an average annual precipitation of 663 mm. The climate is categorized as warm-temperate and fully humid with warm summers (Kottek et al., 2006) i.e. Cfb according to the Köppen-Geiger climate classification.

### 2.2 Experimental design and flux estimation

For the method comparison between GC and LAS, fluxes were determined at two dates (21st September 2023 after winter wheat and tillage, and on 24th March 2024 in the subsequent clover-grass mixture) in plots (Fig. 1) containing two treatments, which differed also in organic fertilization (both treatments were fertilized with 100 kg N ha$^{-1}$ as hair meal pellets; one treatment additionally received green waste compost at 5 t ha$^{-1}$ a$^{-1}$ (dry matter)), replicated four times. This resulted in sixteen sets of comparative data. Gas samples were taken from the chamber by LAS and with syringes and subsequent analysis by GC. In our study, we compared both methods by carrying out simultaneous measurements with closed chambers using the static and dynamic approach, respectively. The chambers were made from non-transparent PVC and fitted with a fan, thermometer, vent (Hutchinson and Livingston, 2001), and a closable opening to prevent pressure pumping during placement. On average, the





soil collars (50 x 50 cm) were installed up to a soil depth of 12 cm, the chamber itself had a dimension of 50 cm x 50 cm x 50 cm. The total volume of the setup, including measurement cells, filters, tubes and chambers depends on the depth of the soil collars, which was determined for each collar individually and was on average 0.1576 m$^3$ (157.56 liters). Tubes were made of

Polytetrafluorethylen (PTFE). The enclosure time of the chambers was 30 minutes and for each chamber measurement, the initial and final temperature (Probe thermometer, *LT-101, TFA Dostmann GmbH & Co. KG, DE*) values were averaged for the flux calculation. Local air pressure data was obtained from a nearby (~200 m distance) weather station *(ATMOS 41, METER Group, Munich, DE)*.

**(a)** **(b)**

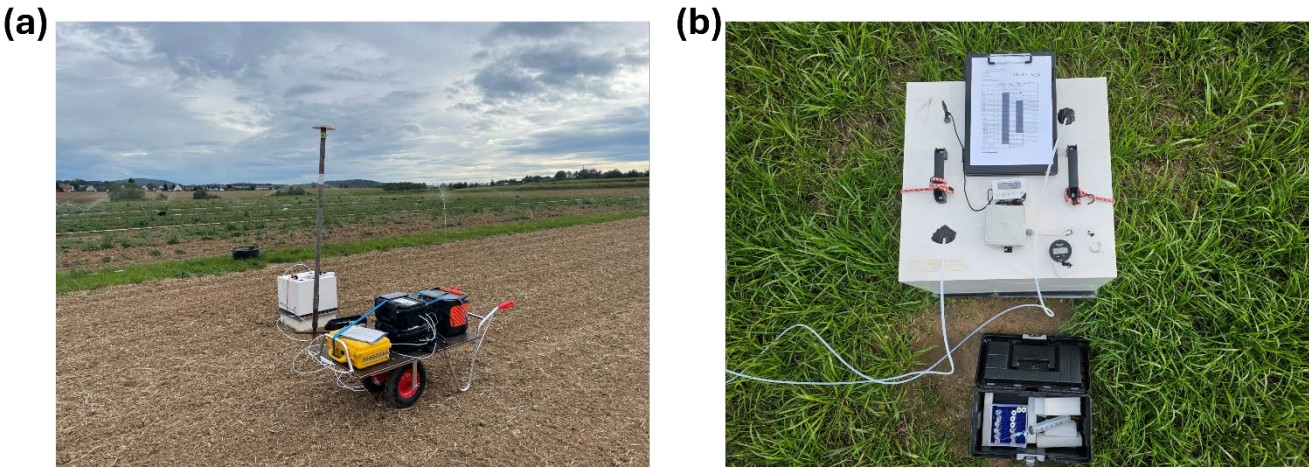

**Figure 1: Field sampling after winter wheat and tillage on 2023-09-21 (a) and in a clover grass mixture on 2024-03-24 (b).**

We analyzed the gases $CO_2$, $N_2O$ and $CH_4$ simultaneously, (i.) in real time on the field with a *MIRA Ultra N$_2$O/CO$_2$* and a *MIRA Ultra Mobile LDS: CH$_4$/C$_2$H$_6$* analyzer using LAS (Direct Absorption) *(AERIS Technologies, Inc., USA)* connected with

a multiplexer PRI-8600D *(Pri-eco Technology Co. LTD, CHN),* and (ii.) by taking gas samples with syringes at six points in time and subsequent analysis by GC. For GC analysis, the samples were transferred to pre-evacuated 12 ml glass vials with grey chlorobutyl rubber septa *(Labco Limited, UK)*. The vials were filled with slight overpressure (~20 ml gas sample squeezed into the 12 ml container). The GC used was a *Bruker Model 450* *(Bruker Corp., USA)* with three separate detectors: Thermal Conductivity Detector (TCD) for $CO_2$, Flame Ionization Detector (FID) for $CH_4$ and Electron Capture Detector (ECD) for

$N_2O$. Before each run, four standard gases (SG) *(DEUSTE Gas Solutions GmbH, DE)* were used, in ascending concentration order, for calibration (see Table A1). In addition, a vial of standard gas three (SG 3) was measured every 43 samples as a control.

Prior to flux calculation, we removed the first and last 30 seconds from the LAS datasets to minimize potential disturbances caused by chamber closure and opening, resulting in a total of 1740 s (i.e. ~1740 data points) for the flux calculation. From

here on, however, this approach is referred to as a 30-minute chamber closure time. In a first step, fluxes were calculated based



on the full 30-minute chamber closure period using two different R packages: *gasfluxes* (version 0.4-4) (Fuss and Hüppi, 2024), which is commonly applied to GC data but not typically used with high-frequency LAS measurements, and *goFlux* (version 0.2.0) (Rheault et al., 2024), which is specifically designed for high-frequency and data incorporates corrections for water vapor dilution (LI-COR, 2023). GC data were processed using *gasfluxes*, while LAS data were analyzed with both

*gasfluxes* and *goFlux*. To reduce the influence of model selection on the results, we applied robust linear regression (provided in the *goFlux* script and accounts for the weighting of outlier data points in flux calculation) to the GC data and standard linear regression to the LAS data. For the GC data, please note that for each of the three gases studied, the robust linear algorithm of the *gasfluxes* script applied robust weighting in only 8 out of 16 flux calculations. The remaining flux calculations did not differ from those obtained using ordinary linear regression. In a second step, we calculated the LAS-based fluxes using the R

package *goFlux* (Rheault et al., 2024). To evaluate whether shorter measurement intervals with the LGA-derived fluxes could still yield results comparable to those from GC measurements, we shortened the LAS dataset to the initial 20 and 10 minutes to simulate shorter chamber enclosure times and recalculated fluxes using the *goFlux* script. This approach to use *goFlux* for LAS data, is supposed to reflect common practice, where shorter chamber measurement durations are often used. These were then compared to GC fluxes based on the full 30-minute measurement period. Subsequently, we implemented a flux selection

procedure to ensure that the best-fitting model was applied for each flux calculation. Specifically, we employed the MDF approach described by Nickerson (2016) (Eq. 3) to determine the kappa max threshold following Hüppi et al. (2018) (Eq. 5). Based on whether the calculated kappa value exceeded or fell below this threshold, either a linear or a non-linear model (Hutchinson and Mosier Regression model, HMR) (Hutchinson and Mosier, 1981) was selected accordingly. As a result, these flux calculation setups facilitated the comparison of seven distinct approaches: (1) GC data, calculated with the *gasfluxes*

package and robust linear regression (GC_gasf_rl), (2) GC data, calculated with the *gasfluxes* package and model selection (GC_gasf), (3) LAS data, calculated with the *gasfluxes* package and linear regression (LAS_gasf_30_l), (4) LAS data, calculated with the *goFlux* package and linear regression (LAS_gof_30_l), (5) LAS data, 30 minute dataset, calculated with the *goFlux* package and model selection (LAS_gof_30), (6) LAS data, 20 minute dataset, calculated with the *goFlux* package and model selection (LAS_gof_20), (7) LAS data, 10 minute dataset, calculated with the *goFlux* package and model selection

(LAS_gof_10).

All flux estimates were multiplied with a flux term:

$$flux.term.gasfluxes = \frac{V\,M\,P}{S\,R\,T} \tag{1}$$

where $V$ is the total chamber volume (m$^3$), $M$ is the molar mass of the measured gas (mol), $P$ is the local atmospheric air pressure (Pa), $S$ is the soil surface covered by the chamber (m$^2$), $R$ is the universal gas constant (8.314 J m$^3$ Pa mol$^{-1}$ K$^{-1}$) and

$T$ is the temperature inside the chamber (K).



Additionally, the *goFlux* package corrects for the dilution effect caused by the increase of water vapor inside the chamber during the measurement (LI-COR, 2023), expressed as follows:

$$flux.term.goflux = \frac{(1-H_2O)\,V\,P}{S\,R\,T} \qquad (2)$$

Please note, that in the *goFlux* script, $V$ is given in liters (L) and therefore $R$ in L kPa $K^{-1} \cdot mol^{-1}$. The term for $M$ is omitted, as fluxes in the script are calculated in mol and $H_2O$ is the water vapor (mol $mol^{-1}$). Subsequently, the values were converted to the same units as in the *gasfluxes* script ($CO_2$ (g $m^{-2}$ $h^{-1}$), $N_2O$ and $CH_4$ (mg$^{-2}$ $h^{-1}$)), using the respective values for $M$.

The methodical detection limit for the measured fluxes was determined based on MDF following the approach of Nickerson (2016). The MDF ($CO_2$ (g $m^{-2}$ $h^{-1}$), $N_2O$ and $CH_4$ (mg$^{-2}$ $h^{-1}$)), was calculated as follows:

$$MDF = \frac{A_A}{t_c\sqrt{\frac{t_c}{p_s}}}\left(\frac{VP}{SRT}\right) \qquad (3)$$

where $A_A$ is the analytical precision of the instrument (ppm), $t_C$ is the closure time of the chamber (h), and $p_S$ is the sampling periodicity (h), whereby $V$ and $R$ again given in in the same units as in Eq. (1). The variables $V$, $P$, and $T$ were averaged over the measurement period covering eight plots (chamber measurements) for each of the two measurement days. The analytical precisions ($A_A$) according to the manufacturer of the LAS analyzer were 200 ppb for $CO_2$, 0.001 ppb for $CH_4$, and 0.0002 ppb for $N_2O$. The sampling frequency was 1 Hz and the analytical accuracy $A_A$ for the two GC runs was calculated following Christiansen et al. (2015b):

$$A_A = 3 \cdot t_{99\%} \cdot SD \qquad (4)$$

where $t_{99\%}$ is the t value at the 99 % confidence interval at df = 4 (4.604) and $SD$ is the standard deviation of five samples standard gas 3 (SG 3, see Table A1) within one run. The average $A_A$ of the two GC runs was 90.6 ppm for $CO_2$, 0.4 ppb for $CH_4$, and 0.1 ppb for $N_2O$.

In methodological reference to Hüppi et al. (2018) the kappa max threshold (k.max ($h^{-1}$)) was calculated as follows:

$$k.max = \frac{linear\ flux}{MDF\ t_C} \qquad (5)$$

On the first day of measurements (2023-09-21), the data set showed gaps for LAS data after 20 minutes in two out of eight plots. For all scatterplots analyses only complete datasets were used. To avoid introducing inconsistencies in the method comparisons based on boxplot representations, the flux values for these plots were calculated based on the available 20-minute





data and copied to in the 30-minute dataset. This decision was further supported by the observation that the differences between
        the 20- and 30-minute measurements were minimal.

## 2.3. Statistical analysis

To evaluate the linear relationship between the analytical approaches, we first calculated Pearson's correlation coefficient.
The deviation of absolute fluxes was quantified using the Root Mean Square Error (RMSE), which measures the average

magnitude of differences between paired observations. To facilitate interpretation and enable comparison across different flux
        magnitudes, we additionally calculated the normalized RMSE (nRMSE) by dividing RMSE by the means across the compared
        observations for each compared data set (five comparisons: see Table 2, 3 and 4). This approach does not assume one method
        as a "true" reference but rather assesses relative agreement between the two methods. The resulting nRMSE was expressed as
        a percentage, allowing for a standardized evaluation of method agreement independent of the absolute flux magnitude.

Additionally, flux values from the seven different approaches were compared using a Kruskal–Wallis to test  for significant
        differences, since the data was not normally distributed.

## 3. Results

The comparison of calculated MDF for $CO_2$, $N_2O$, and $CH_4$ showed that the LAS method yields much lower MDFs than the
GC method for all gases and chamber enclosure durations (Table 1). In the case of LAS, however, flux magnitude sensitivity

decreases with shorter enclosure times, as indicated by increasing MDF values. Overall, $CO_2$ fluxes obtained from the GC and
        LAS methods generally agree well (Fig. 2, Tab. 2). Figure 2a shows GC-derived fluxes (*gasfluxes* script, robust linear
        regression) plotted against GA-derived fluxes for a 30-minute chamber closure time, using the two calculation scripts *gasfluxes*
        (linear regression) and *goFlux* (linear regression). Most data points align closely with the line of equality, indicating a strong
        agreement between GC and LAS, as well as between both LAS-based calculation approaches. Figure 2b presents fluxes based

on flux-model selection derived from GC fluxes (*gasfluxes* script), compared to LAS fluxes with different enclosure durations
        (30, 20 and 10 minutes) calculated using the *goFlux* script that were also selected using the same flux-model selection
        approach. For all $CO_2$-LAS fluxes calculated using the *goFlux* script, the flux selection algorithm applied in a second step
        resulted in HMR model fits for all fluxes. In contrast, flux selection for the GC fluxes (*gasfluxes* script) could only be carried
        out to a limited extent (31 %). This is because HMR models and the corresponding kappa values were available for only a

minority of these fluxes, as the script's internal diagnostics considered the HMR model unsuitable in most cases. The data
        points cluster closely around the line of equality across all closure durations, with high correlation coefficients (r≈1), resulting
        in nRMSE values below 17 % (Table 2). The greatest scatter is observed for the 10-minute duration, indicating lower
        agreement. All fluxes exceeded the MDF and the median of the flux magnitudes of the investigated methodological approaches
        was relatively similar whereas the median of the GC fluxes was in tendency higher (Fig. 2c).






**Table 1: Minimum detectable fluxes (MDF) for $CO_2$, $CH_4$, and $N_2O$, based on the approach by Nickerson (2016), using Gas Chromatography (GC) and mid-infrared laser absorption spectroscopy (LAS). Calculations were performed for a 30-minute chamber closure duration (GC_30 and LAS_30), as well as for shorter durations of 20 and 10 minutes using the LAS (LAS_20 and LAS_10). Data represents averages from two measurement days.**

| Method | $CO_2 \times 10^{-5}$ (g m$^{-2}$ h$^{-1}$) | $N_2O \times 10^{-5}$ (mg m$^{-2}$ h$^{-1}$) | $CH_4 \times 10^{-5}$ (mg m$^{-2}$ h$^{-1}$) |
|---|---|---|---|
| **GC_30** | 940 | 960 | 1,510 |
| **LAS_30** | 1.13 | 1.13 | 2.05 |
| **LAS_20** | 2.04 | 2.04 | 3.72 |
| **LAS_10** | 6.00 | 6.00 | 10 |



Figure 2: Comparison of CO₂ fluxes derived from gas chromatography (GC) and mid-infrared laser absorption spectroscopy (LAS) measurements using different chamber closure durations and calculation approaches. (a) GC derived fluxes for a 30-minute chamber closure duration, calculated using the *gasfluxes* script with robust linear regression (GC_gasf_rl), are plotted against LAS derived fluxes also for a 30-minute closure duration. The LAS fluxes were calculated using either the gasfluxes script (LAS_gasf_30_l) or the *goFlux* script (LAS_gof_30_l), both with linear regression. (b) Model-selected GC fluxes for a 30-minute closure duration using the *gasfluxes* script (GC_gasf_30) are compared to model-selected LAS fluxes using the *goFlux* script for 30, 20, and 10-minute closure durations (LAS_gof_30, LAS_gof_20, LAS_gof_10). In both panels (a) and (b), the black line represents the line of equality, while coloured lines indicate regression lines for the respective approaches. For CO₂, all flux values exceeded the minimum detectable flux (MDF). (c) Boxplots of all calculated CO₂ fluxes across methods and durations (n = 16). For the summary of the mathematical analysis underlying the scatter plot comparisons, see Table 2.





**Table 2: Regression results comparing CO$_2$ fluxes derived from Gas Chromatography (GC) and mid-infrared laser absorption spectroscopy (LAS) with varying chamber closure times and calculation scripts. Displayed are the correlation coefficient (r), regression equation, root mean square error (RMSE), and normalized RMSE (nRMSE).**

| Method | r | Equation | RMSE (g m$^{-2}$ h$^{-1}$) | nRMSE (%) |
|---|---|---|---|---|
| LAS_gasf_30_l vs. GC_gasf_rl | 1.00 | y = 0.09 + 0.94x | 0.057 | 5.79 |
| LAS_gof_30_l vs GC_gasf_rl | 1.00 | y = 0.08 + 0.96x | 0.057 | 5.80 |
| LAS_gof_30 vs. GC_gasf | 1.00 | y = 0.07 + 1.01x | 0.086 | 8.51 |
| LAS_gof_20 vs. GC_gasf | 1.00 | y = 0.07 + 1.02x | 0.100 | 9.38 |
| LAS_gog_10 vs. GC_gasf | 0.97 | y = 0.08 + 1.03x | 0.176 | 16.70 |

For N$_2$O, a generally good agreement was also observed between the flux values derived from GC and LAS flux estimation (Fig. 3, Table 3). This was observed both, without (Fig. 3a) and with the flux selection procedure (Fig. 3b). For N$_2$O, flux selection for LAS fluxes calculated with the *goFlux* script always resulted in HMR models. In contrast, model selection for GC fluxes using the *gasfluxes* script was only possible for a small subset of fluxes (18 %) due to the absence of HMR models and corresponding kappa values (HMR diagnostics). Notably, obtaining HMR models with GC fluxes was only possible on

the first measurement day, when an N$_2$O emission pulse occurred. The values cover a wide range of flux magnitudes, including very low and relatively high values. The data points align closely along the line of equality, and only a few GC measurements fell below the MDF (low flux range), whereas all LAS fluxes (even small negative fluxes) were above the MDF. The median flux magnitudes were relatively similar overall, with the GC fluxes tending to be higher, as was also observed for CO$_2$. The nRMSE varied between 14.63 % and 24.64 %, with the highest values corresponding to the shortest LAS closure time (10

minutes), as also observed for CO$_2$.






**Figure 3: Comparison of N₂O fluxes derived from gas chromatography (GC) mid-infrared laser absorption spectroscopy (LAS) measurements using different chamber closure durations and calculation approaches. (a) GC derived fluxes for a 30-minute chamber closure duration, calculated using the *gasfluxes* script with robust linear regression (GC_gasf_rl), are plotted against LAS derived fluxes also for a 30-minute closure duration. The LAS fluxes were calculated using either the gasfluxes script (LAS_gasf_30_l) or the *goFlux* script (LAS_gof_30_l), both with linear regression. (b) Model-selected GC fluxes for a 30-minute closure duration using the *gasfluxes* script (GC_gasf_30) are compared to model-selected LAS fluxes using the *goFlux* script for 30, 20, and 10-minute closure durations (LAS_gof_30, LAS_gof_20, LAS_gof_10). In both panels (a) and (b), the black line represents the line of equality, while coloured lines indicate regression lines for the respective approaches. For N₂O, only higher fluxes exceeded the minimum detectable flux (MDF) (points filled in black), whereas all LAS flux values exceeded the MDF (points filled in grey). (c) Boxplots of all calculated N₂O fluxes across methods and durations, n = 16. durations (n = 16). For the summary of the mathematical analysis underlying the scatter plot comparisons, see Table 3.**






| Method | r | Equation | RMSE (mg m$^{-2}$ h$^{-1}$) | nRMSE (%) |
|---|---|---|---|---|
| **LAS_gasf_30_l vs. GC_gasf_rl** | 1.00 | y = 0 + 0.98x | 0.010 | 15.01 |
| **LAS_gof_30_l vs GC_gasf_rl** | 1.00 | y = 0 + 1x | 0.010 | 14.63 |
| **LAS_gof_30 vs. GC_gasf** | 0.99 | y = 0 + 1.02x | 0.012 | 18.21 |
| **LAS_gof_20 vs. GC_gasf** | 0.99 | y = 0 + 1.04x | 0.013 | 17.01 |
| **LAS_gog_10 vs. GC_gasf** | 0.99 | y = 0.01 + 1.03x | 0.018 | 24.64 |

Figure 4 illustrates the comparison of CH$_4$ fluxes between GC and LAS methods without (Fig. 4a) and with model selection (Fig. 4b). In contrast to CO$_2$ and N$_2$O, the data points displayed substantial scatter, and the agreement with the line of equality was visible weaker (compare to Fig. 2 and 3) and the nRMSE values remarkable higher (compare to Tables 2 and 3). For CH$_4$, flux selection for LAS fluxes calculated with the *goFlux* script always resulted in HMR models. In contrast, model selection for GC fluxes using the *gasfluxes* script was only possible for a minority of the fluxes (18 %) due to the absence HMR and kappa values, as a result of the script's internal HMR diagnostics. Despite some spread, the flux value distributions across GC and LAS measurements appear broadly similar (Fig. 4c), without significant differences between the approaches, although the median of the GC data tended to be higher (more negative). All GC fluxes were below the MDF, whereas nearly all LAS fluxes except for one value with 10-minute enclosure time were above it. The regression analysis confirmed the visual impression of lower consistency. The correlation coefficients (r) remained below 0.54 in all cases and the nRMSE values ranged between 88.42 % and 94.54 %, indicating relatively large relative deviations (Tab. 4).







**Figure 4: Comparison of CH$_4$ fluxes derived from gas chromatography (GC) and mid-infrared laser absorption spectroscopy (LAS) measurements using different chamber closure durations and calculation approaches. (a) GC derived fluxes for a 30-minute chamber closure duration, calculated using the *gasfluxes* script with robust linear regression (GC_gasf_rl), are plotted against LAS derived fluxes also for a 30-minute closure duration. The LAS fluxes were calculated using either the gasfluxes script (LAS_gasf_30_l) or the *goFlux* script (LAS_gof_30_l), both with linear regression. (b) Model-selected GC fluxes for a 30-minute closure duration using the *gasfluxes* script (GC_gasf_30) are compared to model-selected LAS fluxes using the *goFlux* script for 30, 20, and 10-minute closure durations (LAS_gof_30, LAS_gof_20, LAS_gof_10). In both panels (a) and (b), the black line represents the line of equality, while coloured lines indicate regression lines for the respective approaches. For CH$_4$, no GC flux values exceeded the minimum detectable flux (MDF), while almost all LAS flux values did (points filled in grey). Only one LAS flux fell below the MDF (indicated by an empty fill, located in the top right corner). (c) Boxplots of all calculated CH$_4$ fluxes across methods and durations, n = 16. For the summary of the mathematical analysis underlying the scatter plot comparisons, see Table 4.**



**Table 4: Regression results comparing CH$_4$ fluxes derived from GC and LAS methods with varying chamber closure times and calculation scripts. Displayed are the correlation coefficient (r), regression equation, root mean square error (RMSE), and normalized RMSE (nRMSE).**

| Method | r | Equation | RMSE (mg m$^{-2}$ h$^{-1}$) | nRMSE (%) |
|---|---|---|---|---|
| LAS_gasf_30_l vs. GC_gasf_rl | 0.54 | y = -0.01 + 0.55x | 0.027 | 89.40 |
| LAS_gof_30_l vs GC_gasf_rl | 0.54 | y = -0.01 + 0.56x | 0.027 | 89.34 |
| LAS_gof_30 vs. GC_gasf | 0.52 | y = -0.01 + 0.51x | 0.028 | 91.36 |
| LAS_gof_20 vs. GC_gasf | 0.51 | y = -0.01 + 0.56x | 0.027 | 88.42 |
| LAS_gog_10 vs. GC_gasf | 0.46 | y = -0.01 + 0.55x | 0.028 | 94.54 |


## 4. Discussion

Overall, the measurement setup was suitable for a method comparison, as it enabled the detection of both high and low flux magnitudes across all three investigated gases. While possible treatment effects (two treatments) could have offered further

insights, neither were statistically significant in the dataset and were therefore not investigated in detail.  The calculation of the MDF according to Nickerson (2016) revealed marked differences between the GC and LAS methods, which can be attributed to substantial disparities in analytical precision. As a result, the LAS method was able to detect significantly lower fluxes than the GC method, representing a clear analytical advantage. The LAS-method's high sensitivity allowed for the detection of fluxes that would have remained below the detection threshold of the GC and would therefore have been classified

as not significantly different from zero. Consequently, the enclosure time could be reduced substantially (e.g., to 10 minutes), as also recommended in other studies (Brümmer et al., 2017), without leading to a strong increase in the MDF, offering additional methodological flexibility and potential reduction of measurement duration-induced disturbances.

For CO$_2$, a high level of agreement was observed between the GC and LAS data pairs across all chamber durations and both scripts (*gasfluxes* and *goFlux*). The MDF was exceeded in all cases for both LAS and GC, facilitating comparability.

Absolute flux values did not differ significantly between approaches, and the nRMSE remained low across all comparisons (max. 17 %, see Table 2). The regression slopes were close to 1, and intercepts near zero, indicating that both systems reliably captured actual CO$_2$ fluxes under field conditions. In line with Cowan et al. (2025), this shows that the traditional and still widely employed closed static chamber method is not necessarily inferior to closed dynamic chamber approaches in situations where the measured fluxes are considerably larger than the analytical uncertainty. Depending on the study site and the research

question, the choice between GC and LAS can then rather be guided by logistical and budgetary constraints.



For $N_2O$, the comparison revealed clear differences between the two measurement days. On the first day, flux magnitudes were relatively high, likely caused by a recent tillage event (initial shallow tillage was performed with a rotavator (~4 cm depth), followed by deeper rotary tillage with a rotary tiller), and MDFs were exceeded by both methods. On the second day,

however, flux magnitudes were low and, in some cases, negative. The capture of this $N_2O$ pulse, along with the very low to negative fluxes, demonstrated a strong agreement between methods, with only minimal error for the 30-min enclosure time (~15 %,) and a tolerable deviation for 10-min enclosure time (25 %) (see Table 3). Negative fluxes were consistently detected and quantified by the LAS method, whereas the GC measurements for these fluxes fell below the MDF, resulting in no significant difference from zero flux. This demonstrates the superior ability of LAS to measure very low $N_2O$ fluxes and even

small net uptake events. Such negative fluxes are commonly attributed to the final step of denitrification, where $N_2O$ is reduced to $N_2$ (Cavigelli and Robertson, 2001; Glatzel and Stahr, 2001; Butterbach-Bahl et al., 2002). Although these processes are well documented, they are frequently underestimated or omitted in flux datasets, as negative net fluxes often fall below detection thresholds. Chapuis-Lardy et al. (2007) highlight that this omission may lead to critical misinterpretations of the global $N_2O$ budget. The high analytical sensitivity of the LAS method could help to better integrate $N_2O$ sinks, but also very

low $N_2O$ emissions into future flux assessments (Cowan et al., 2025; Triches et al., in review, 2025).

The results for $CH_4$ showed considerable divergence. All measured LAS-derived $CH_4$ fluxes were negative, as were almost all GC-derived $CH_4$ fluxes, as expected for well-aerated upland soils where $CH_4$ oxidation by methanotrophic bacteria predominates (Hütsch, 1998; Powlson et al., 2014). While the LAS measurements consistently exceeded the MDF, the GC fluxes did not, likely resulting in a higher degree of uncertainty in the GC dataset and thus a lower ability to detect potential

treatment effects in $CH_4$ consumption rates. This discrepancy may have led to inflated variability in the GC results, particularly at low flux values (near zero). Even in the low concentration range, the LAS provided significant results (flux > MDF) for nearly all fluxes (with only one exception, with a 10-minute chamber enclosure time), highlighting its advantage in terms of flux sensitivity. Despite these differences and the low agreement of measured fluxes, the mean $CH_4$ fluxes did not differ significantly between methods (see Fig. 4c), suggesting that any potential effects induced by treatment factors might be

captured in a similar manner. However, the low correlation ($r \approx 0.5$) and high nRMSE up to 95 % indicate that more extensive datasets for examples with more values exceeding MDF for GC fluxes are needed to validate this assumption. However, within this study, $CH_4$ oxidation rates were in part comparatively high: $CH_4$ oxidation rates in aerated soils rarely exceed values of 0.1 mg m$^{-2}$ h$^{-1}$ (Le Mer and Roger, 2001), but in our case, we observed rates up to 0.8 mg m$^{-2}$ h$^{-1}$. This leads us to assume that more extensive data sets of simultaneous chamber measurements would not necessarily provide further clarity. For further

investigations, reducing chamber volume may represent the most effective measure to minimize MDF. However, in field campaigns, this approach is often not feasible, as it is often of interest to include plants within the chamber and also to level out spatial variability over capturing larger soil surfaces (e.g. heterogeneous distribution of solid manure and composts (Krauss et al., 2017)). For verification of the assumption, that treatment effects can be captured in a similar manner, both with GC and LAS, measuring a time series with different treatments and calculating cumulative fluxes is advisable. In any case, for well-

aerated soils using relatively large chamber setups, $CH_4$ fluxes are generally expected to fall below the MDF for the GC



method. Consequently, under these conditions, the measurements obtained using LAS may be considered more reliable for investigating $CH_4$ uptake and treatment effects on $CH_4$ consumption in arable soils.

The application of the *goFlux* script, including correction for water vapor inside the chamber during the measurement (dilution
effect) (LI-COR, 2023), had only marginal effects on the absolute flux magnitudes and on the agreement between methods. This suggests that, in the present dataset, absolute humidity had little impact on flux calculations, despite noticeable differences between measurement days. This is in contrast to Kong et al. (2025), who found a significant effect of water vapor corrections on $N_2O$ fluxes, especially below 50 µg $N_2O$-N m$^{-2}$ h$^{-1}$ (0.079 mg $N_2O$ m$^{-2}$ h$^{-1}$, own conversion), contributing to discrepancies in cumulated $N_2O$ emission estimates in a similar chamber-based method comparison campaign on Danish arable soils. The
flux selection approach was applied according to Hüppi et al. (2018) (LAS and GC). While HMR models were selected in all cases for the LAS fluxes, the *gasfluxes* script only allowed HMR calculations for a minority of the GC fluxes ($CO_2$: 31 %, $N_2O$ and $CH_4$: 19%), as most did not meet the criteria defined by the script internal HMR diagnostics. On the first measurement day, when a high $N_2O$ emission peak occurred, 3 out of 8 fluxes could be calculated using the HMR model, while on the second measurement day, however, none could be fitted with HMR. This illustrates that applying HMR calculations to GC-based
measurements is often problematic, particularly at very low flux magnitudes near the MDF. It remains unclear whether this is primarily due to the limited number of measurement time points (only six per chamber) or the lower analytical precision of the method, or a combination of both. Although the *gasfluxes* script's internal HMR diagnostics allowed significantly more HMR model calculations for LAS compared to GC fluxes, it somewhat surprisingly still did not provide HMR models for all LAS fluxes, unlike the *goFlux* script. Nevertheless, when comparing the two methods (GC and LAS) with and without flux
selection, it became apparent that there was only a small influence of the model selection procedure on our results and it did not significantly affect the relationship between the two methods over the 30-minute closure periods. In contrast, the shorter chamber closure durations had a more pronounced effect on the flux estimates. The relative error for $CO_2$ and $N_2O$ was highest with the LAS when using a 10-minute closure time, compared to the GC fluxes, which were always based on a 30-minute enclosure. However, even in this case, the error remained within a moderate range for these two gases. Also, for $CH_4$, reducing
the enclosure time to 20 and 10 minutes had only a minimal effect on the calculated LAS-derived fluxes. In addition to the only slightly reduced MDF resulting from the shortened chamber enclosure durations (see above), this further highlights the LAS method's suitability in terms of operational practicability and the reduction of measurement duration-induced disturbance of the gas concentration gradient and hence gas flux dynamics between soil and chamber atmospheres.

Taken together, although no treatment effects were explicitly tested in this study, both GC and LAS approaches resulted in comparable flux magnitudes for $CO_2$ and $N_2O$, suggesting that both methods would very likely provide consistent results when assessing potential treatment effects. For $CH_4$, flux magnitudes also did not differ significantly between methods; however, this apparent agreement is subject to considerably greater uncertainty, as indicated by the scatterplot analysis and the fact that GC fluxes were generally below the MDF and thus not significantly different from zero flux. However, for all three gases, a



pattern of slightly higher fluxes, measured by GC, was observed. The LAS offers several technical and analytical advantages: its low MDF allows for the detection of very small but significant $CH_4$ and $N_2O$ fluxes; it permits shorter enclosure times without substantial increases in uncertainty; and it offers real-time measurements with minimal infrastructure requirements. These features enable faster, more flexible sampling and improve the resolution of short-term flux dynamics, such as during $N_2O$ pulse-emission events. At the same time, practical limitations of the LAS include high acquisition costs, increased

vulnerability to environmental influences (e.g. dust, moisture, high temperature), and maintenance demands, which may affect measurement logistics under field conditions.

## 5. Conclusion

Biotic greenhouse gas (GHG) fluxes are often measured using the closed chamber method, but the choice of analytical instrumentation influences flux estimations and reliability of the results. In this study, we compared gas chromatography (GC)

with mid-infrared laser absorption spectroscopy (LAS) based on simultaneous chamber measurements and derived the following conclusions:

–    The measurement days proved to be suitable for the comparison, as they covered a wide flux range across the investigated gases ($CO_2$, $CH_4$, and $N_2O$) from well-aerated upland soils.

–    For $CO_2$ fluxes, a very high level of agreement between the methods was observed. All fluxes were well above the minimum detectable flux (MDF), and deviations were low (nRMSE < 17 %), confirming the reliability of both methods for measuring $CO_2$.

–    For $N_2O$, the methods showed strong agreement across both measurement days, with high correlation coefficients (r ~ 1) and low deviations (nRMSE < 25 %). For low fluxes GC-derived fluxes fell below the MDF and could

not be distinguished from zero flux.

–    In the case of $CH_4$, the agreement between methods was poor (nRMSE up to 95 %). Despite very high $CH_4$ oxidation rates, none of the GC measurements exceeded the MDF. In contrast, almost all values measured by LAS exceeded the MDF, which likely contributed to the weak correlation between the two methods $CH_4$ fluxes.

–    A central advantage of the LAS method lies in its considerably lower MDF in combination with the higher

measurement frequency (~ 1 Hz), which enables the detection of statistically significant fluxes (flux > MDF) even at a very low flux range. This is especially relevant for $CH_4$ and $N_2O$, where small flux rates occur and uptake of either gases into the soil-plant system might remain undetected.

–    The ability to visualize and validate measurements in real time adds to the practical benefits of LAS, offering increased flexibility in field campaigns. However, high acquisition costs and sensitivity to dust and moisture

remain important limitations that must be considered when choosing the appropriate measurement technology.



    –    Chamber closure time influenced the variability of LAS-derived fluxes, with shorter durations leading to higher measurement uncertainty. However, no statistically significant differences were observed in the absolute flux values between the different closure times of 30, 20 and 10 minutes. This confirms that shorter chamber closure times are suitable for field measurements with LAS, offering more flexibility without compromising result validity.

Overall, this pilot study provides a comprehensive comparison of two widely used analytical techniques for chamber-based GHG flux measurements. Given the high agreement observed between methods for $CO_2$ and $N_2O$, we conclude that LAS is a valid and reliable alternative to the established GC approach for these gases. For quantifying $CH_4$ uptake rates in aerated soils, there is considerable uncertainty regarding the consistency between the two methods; however, this is very likely due to the high MDF of the GC method. In conclusion, the LAS's analytical sensitivity and operational flexibility clearly supports its broader application in trace gas research, especially for gaining new insights into the natural variability of low soil GHG fluxes which are masked by instrumental noise of the traditional closed static chamber method.

**Appendix A: Details of standard gases for gas chromatography measurements**

**Table A1: List of (non-isotope) standards used for testing and calibration[1].**

| Standard gas (SG) | $CO_2$ | | $CH_4$ | | $N_2O$ | | $O_2$ | |
|---|---|---|---|---|---|---|---|---|
| | Conc. (ppm) | Relative error (±%) | Conc. (ppm) | Relative error (±%) | Conc. (ppb) | Relative error (±%) | Conc. (vol%) | Relative error (±%) |
| **SG1** | 304 | 1 | 1.02 | 2 | 248.4 | 3 | 19.01 | 0.5 |
| **SG2** | 402.3 | 0.5 | 1.81 | 2 | 321.3 | 3 | 20.97 | 0.5 |
| **SG3** | 1509.2 | 0.5 | 5.02 | 2 | 2010 | 3 | 15 | 0.5 |
| **SG4** | 3999.6 | 0.5 | 20.9 | 2 | 15100 | 2 | 10 | 0.5 |

[1]According to analysis certificate of the manufacturer (*DEUSTE Steininger GmbH, DE*).



**Data availability**

The processed data that support our findings are included in the supplementary material. For more detailed information, the input files for the flux calculations as well as the output from the calculation scripts are openly available in the repository Zenodo at https://doi.org/10.5281/zenodo.15674498 (Aumer et al., 2025).

**Competing interests**

We have no competing interests to declare.

**Author contributions**

WA and MM contributed equally and took lead in writing the manuscript and analysing the data. CE provided GC raw data, CMG and CK were in charge of conceptualization. CBi provided weather data. AG, CBi, CBr, CE, CK, CMG, MRF, TKDW supported us in writing the manuscript and all authors approved the final version.

**Acknowledgements**

The long-term experiment (AKHWA project, URL: www.akhwa.de) and the laser gas analyzers were funded by the Hessian Ministry for Agriculture, Environment, Viticulture, Forestry, Hunting and Homeland Affairs. Furthermore, we acknowledge the valuable contributions of all field technicians and student assistants involved in the long-term field experiment where this investigation took place.

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
