# Peer review of "Technical note: Gas Chromatography vs. Mid-Infrared Laser Absorption Spectroscopy: A comparison of methods for measuring greenhouse gas fluxes from arable soils"

_EGUsphere, 2025_

## Referee Comment (RC1)

**Review on the technical note: Gas chromatography vs. mid-infrared laser absorption spectroscopy: A comparison of methods for measuring greenhouse gas fluxes from arable soils, submitted to the EGUsphere Preprint Repository.**

The manuscript presents comparative trace gas measurements ($N_2O$, $CO_2$, and $CH_4$) using a GC and a MIR laser system. Both systems are commercially available. The measurements were conducted on two dates in two different treatments on a field under organic farming. These two dates were very well chosen because they represented a wide concentration range, at least for $N_2O$ and $CO_2$. The flux rates calculated using different algorithms showed a very high degree of agreement between GC and laser data for both $CO_2$ and $N_2O$. In contrast, the correlation between GC and laser measurements was significantly poorer for the $CH_4$ fluxes, which was probably a result of low $CH_4$ fluxes and an unfavorable ratio of volume to emitting surface area of the chamber system.

To my best knowledge, the laser used is, apart from an ICL laser described by Stiefvater et al. (2023), the only device with a low weight that can be carried on the back, for example, which may be useful for practical application in the field.

The manuscript is well written, the calculations and statistical methods are correct in my opinion, and the measurements were conducted with technical skill.

The comparison of the two systems is particularly interesting for the greenhouse gas community because the laser used for $CO_2$ and $N_2O$ measurements is currently experiencing strong sales growth in Europe among institutions that have previously worked with conventional GC methods. For this reason, the data is worth publishing!

I was uncertain when evaluating the manuscript. On the one hand, all of my comments are easy to implement, but on the other hand, there were several comments, so I ultimately decided on a **major revision**.

**Following are my comments and suggestions**

*When the authors' original text has been used following, it is indicated in italics!*

Title

The title is somewhat misleading. I initially expected that, for example, different detectors or different device settings would be used for GC or that several laser systems would be employed. Ultimately, two commercially available systems were compared without testing different device settings. Perhaps the authors could include this in the title, stating that two systems were compared and not just two detection methods.

Introduction

Line 43: Delete „*with syringes*" because there are also other collection systems.

l. 53-55: Most columns used for trace gas detection in GC separate $CO_2$ from $N_2O$ (e.g. porapack Q, haysep, etc.). How explained Zheng et al. (2008) this effect of $CO_2$ on the detector response for $N_2O$?

l. 68: Chamber size is somehow too imprecise, better use the ratio of the chamber volume to the emitting surface area (effective chamber height)

l. 66-67: Please insert some citations!

l. 71: I would suggest to use something like: ….. world-wide very common GC method with a $^{63}Ni$ ECD…..

l. 80: The following text has somehow found its' way into the manuscript, remove it: Klicken oder tippen Sie hier, um Text einzugeben

l. 91: When were the fertilizers applied relative to the measurements?

Chapter 2.2.: Flux estimation: Given the amount and quality of data that a laser provides, the use of a generalized additive model may be more appropriate than the use of a parametric model (see Themistokleous et al. 2024, https://doi.org/10.1111/ejss.13560). Insert this information.

l. 93: State the pump rate of the laser

l. 93: Replace „*with syringes*" by „or transferred to evacuated exetainers using syringes"!

l. 96: What was the point of the closable opening when the chambers already were equipped with a vent?

l. 97: Why have the authors used such high chambers? Though there is no optimum effective chamber height (V / A), commonly used chambers have an effective chamber height around 0.2 m, the chambers used for this investigation are in the range of 0.6 m! Please discuss this aspect more intense in the discussion section because it directly affects your results on MDF etc.!

l. 108/109: The lasers are not really well described. Don`t assume every reader is familiar with the laser technique, so describe the principle briefly! Add further information on your devices such as the cell volume and the volume of the tubings, were the cells heated or not?

l. 110: Were the samples taken at equidistant intervals? If so, please specify the intervals!

l. 112: *The vials were filled with slight overpressure (~20 ml gas sample squeezed into the 12 ml container)*. Why? To load sample loops etc.? Explain briefly, don`t assume every reader (including me) knows the functioning of a Bruker autosampler!

l. 113-116: Please insert more information on GC settings that are of relevance for the sensitivity of the ECD towards $N_2O$, like e.g. the detector temperature!

l. 130: Explain abbreviation LGA

l. 192: Explain abbreviation GA

Table 1: Data for $CO_2$ and $N_2O$ are the same for the laser measurements, is that a copying error?

Table 1: Values in g (for $CO_2$) and mg (for $N_2O$) are difficult to compare with literature, present them in more common units as mg $CO_2$ or as µg $N_2O$ and $CH_4$

l. 163: The analytical precision for $N_2O$ is given with 0,0002 ppb. In the Allan plot of the data sheet provided for the MIRA LAS the precision is <100 ppt. This value was also provided by the AERIS company, please correct the value and also check for $CO_2$ and $CH_4$!

Figure 3c: Check Y axis labelling, GC should be wrong, right?

l. 288/289: …. *across all three investigated gases*…… There aren`t any high $CH_4$ fluxes presented, please correct the sentence accordingly.

l.289/290: …. *could have offered further insights* …. Unclear, insights in what?

l. 326: *Even in the low concentration range*…. Should be replaced by something like: … under conditions with only low changes in $CH_4$ concentrations ….

l. 331: *However, within this study, $CH_4$ oxidation rates were in part comparatively high: $CH_4$ oxidation rates in aerated soils rarely exceed values of 0.1 mg $m^{-2}$ $h^{-1}$ (Le Mer and Roger, 2001), but in our case, we observed rates up to 0.8 mg $m^{-2}$ $h^{-1}$.* Figure 4 shows something different: $CH_4$ fluxes ranged between +0.03 and -0.08 mg $m^{-2}$ $h^{-1}$! This range is common for well aerated soils and definitely not exceptional high!

L. 334: *For further investigations, reducing chamber volume may represent the most effective measure to minimize MDF. However, in field campaigns, this approach is often not feasible, as it is often of interest to include plants within the chamber and also to level out spatial variability over capturing larger soil surfaces (e.g. heterogeneous distribution of solid manure and composts (Krauss et al., 2017)).* I agree with the first part of this sentence and would like to encourage the authors to discuss their chamber design concerning the high effective chamber height a little bit more in detail, also with respect to low $CH_4$ fluxes! The argument of growing plants in the chambers justifies the use of chambers with high effective chamber heights. However, at both points in time during the investigation, lower chambers would have been desirable in terms of minor changes in the $N_2O$ and $CH_4$ concentrations of the chambers' atmosphere. Further, I don`t understand the argument with the spatial variability. In order to minimize large spatial variability, the area covered by the chambers must be larger, but the ratio of chamber volume to emitting area can still be maintained!

l. 344: *The application of the goFlux script, including correction for water vapor inside the chamber during the measurement (dilution effect) (LI-COR, 2023),* …. As far as I know, the AERIS Ultra already measures water vapor and corrects trace gas concentrations accordingly. Why is there a further correction needed?

l. 347: … *This is in contrast to Kong et al. (2025), who found a significant effect of water vapor corrections on $N_2O$ fluxes, especially below 50 µg $N_2O$-N $m^{-2}$ $h^{-1}$*….. Please specify the

detection mode in Kong et al.! Did they also use a MIR laser spectroscopy? And if so, what do the authors expect as a reason for this discrepancy?

l. 371: *However, for all three gases, a pattern of slightly higher fluxes, measured by GC, was observed.* Unfortunately, the authors don't provide any reason for that phenomenon. Are there any ideas?

l. 379: *At the same time, practical limitations of the LAS include high acquisition costs...* Costs for the $CO_2$/$N_2O$ laser are approximately 45,000 US $ and this is by far lower than the costs for a GC. If you also consider that the laser system does not require any sample preparation (evacuating gas vials), no costs for GC operating gases, and no actual GC measurement, the costs for the laser will definitely be lower than for the GC!